# A High Bandwidth-Power Efficiency, Low THD[2,3] Driver Amplifier with Dual-Loop Active Frequency Compensation for High-Speed Applications

**Ximing Fu** [1] , **Kamal El-Sankary** [1] **and Yadong Yin** [2,*]

1  Department of Electrical and Computer Engineering, Dalhousie University, Halifax, NS B3H 4R2, Canada; xm954365@dal.ca (X.F.); kamal.el-sankary@dal.ca (K.E.-S.)
2  School of Physics and Information Engineering, Fuzhou University, Fuzhou 350002, China
*  Correspondence: yinyadong@fzu.edu.cn; Tel.: +86-591-22865132

**Abstract:** This paper presents a driver amplifier with high bandwidth-power efficiency, high capacitor-driving capacity, and low total harmonic distortion (THD). One complementary differential pair composed of self-cascode transistors is incorporated to obtain a full input voltage swing. Flipped voltage follower (FVF) buffers are applied as second stage to drive the last class-AB output stage. Moreover, a dual-loop active-feedback frequency compensation (DLAFC) is presented, which can stabilize the proposed multistage amplifier and keep the dominant pole on high frequency to obtain high-frequency total harmonic distortion (THD) suppression. To achieve a low-frequency phase margin protection (PMP), one left half-plane (LHP) zero is introduced to compensate for the non-dominant pole caused by the load capacitor. Meanwhile, two high-frequency LHP zeros are injected to achieve high-frequency phase margin boosting (PMB) and reduce the amplifier's settling time and integration area. This proposed amplifier is implemented in a standard DBH 0.18 μm 5 V CMOS process, and it achieves over 115-dB DC gain, 150–300 MHz GBW under 0–100 p load capacitors, ultra-high THD[2,3] suppression ranges from 100 kHz to 10 MHz under 1–2 V output swing, and over 250 V/μs average slew rate, by only dissipating 12.5 mW at 5 V power supply.

**Keywords:** cascode compensation; current buffers (CBs); multistage amplifiers; nested Miller compensation (NMC); reverse NMC (RNMC)

## 1. Introduction

Driving amplifiers with high speed, low distortion, and strong capacitive driving ability are widely used in many industrial and measurement applications, such as high resolution (>16-bit) analog-to-digital converters (ADCs) [1–5], liquid crystal display (LCD) [6–11], etc. In high-resolution ADCs, the challenge is that the driver amplifiers should be able to drive the sampling capacitors with a wide range of capacitance while maintaining the distortion and noise levels lower than those of the ADCs. Driving amplifiers also determine the speed, resolution, voltage swing, and power dissipation of the LCD drivers [7–10]. LCD panels on multimedia products have become larger with higher definition, and their color quality requires more accuracy. With the advancement of liquid crystal display (LCD), there is a large demand for developing driving amplifiers in high-resolution and high color-depth driver ICs [6–11].

Multistage amplifier design has proven to be a very effective approach in driver amplifier design. As known, multistage amplifiers can be very power-efficient in driving a large capacitive load and generally achieve a high bandwidth compared with single-stage amplifiers [12,13]. The common challenge of multistage amplifiers is to achieve stability under a wide range of load and high power-bandwidth efficiency. A three-stage amplifier has at least three high-impedance nodes; therefore, each will contribute a pole in the frequency domain. Without a dedicated frequency compensation scheme, these

poles will be located at low frequency, leading to stability problems. The simple pole-splitting effect can help amplifiers achieve stability by shrinking the dominant pole to the lower frequency; however, it will reduce the amplifier's bandwidth. In recent years, various frequency compensation techniques have been developed to ensure system stability, such as damping-factor-control frequency compensation (DFCFC) [12], active-feedback frequency-compensation (AFFC) [13], and nested/reversed Miller compensation (NMC, RNMC) [14,15]. Most of these structures are derived from the nested Miller compensation (NMC). The major challenge of NMC is to maintain the balance between the complex-pole frequency and the quality factor Q. The major contribution of the proposed DLAFC is to optimize the complex conjugate poles by injecting two left half-plane (LHP) zeros before the complex-pole frequency. Meanwhile, one extra LHP zero is generated to compensate for the negative phase shift caused by the nondominant pole, which improves the phase margin protection. Finally, the proposed amplifier will result in high gain-bandwidth products (GBW), high total harmonic distortion (THD) suppression, and fast settling time. This paper is organized as follows. Section 2 will give a detailed analysis of architecture selection and the proposed frequency compensation technique for the proposed amplifier. In Section 3, the simulations based on the proposed design and the comparison table of state-of-art will be presented to validate the proposed design principles. In Section 4, the conclusion is drawn.

## 2. Multistage Amplifier Architecture

### 2.1. Zinput Stage and Class-AB Output Stage

The circuit's implementation, shown in Figure 1, is proposed to prove the theoretical analysis and proposed amplifier design strategy. In this design, a complementary CMOS input stage is adopted for rail–rail common-mode level. As Figure 2a shows, the input stage consists of an nMOS differential pair and a pMOS differential pair in parallel. Due to the input voltage swing limitations, a traditional differential pair is not capable of processing signals with rail-to-rail common-mode levels. Each differential pair in the complementary stage can keep operating even if the input common-mode voltage is close to the positive (negative) power supply rail. When the input common-mode voltage is in the mid-supply range, both pairs are active. Therefore, no matter where the input common-mode voltage is between the two supply rails, the input stage will contribute transconductance (gmp+gmn, as shown in Figure 2b). It is worth mentioning that the first output stage uses a triple cascode to achieve high DC gain rather than a regulated loop for gain boosting because the regulated loop will introduce more extra poles within the bandwidth, which will create extra unnecessary difficulty and limitation on frequency compensation.

To further increase the input stage output impedance, the composite cascode structures, also referred to as the self-cascode structure, are applied to replace normal NMOS and PMOS transistors in the differential pairs. As shown in Figure 2c, two transistors M1 and M2 are stacked while their gates are connected. The ratio of channel width and length ($W/L$) of the transistor M1 is set larger than M2's; consequently, M1 will work in saturation region while M2 operates in triode region since the drain-source voltage of M1 in Figure 2b, $V_{S2}$-$V_S$, is smaller than its overdrive voltage. Therefore, the output resistance of the self-cascode transistors can be calculated as:

$$\mathrm{r_o} = g_{m1} r_{o2} r_{o1} + r_{o1} + r_{o2} \tag{1}$$

wherein $g_{m2}$ and $r_{o2}$ are the transconductance and drain-source resistance of M2, respectively, $r_{o1}$ is M1's drain-source resistance. Since M1 is in the saturation region while M2 is in the triode region, if the body effect of M2 is ignored, we can get that:

$$\mathrm{I_{M1}} \cong \frac{1}{2}\beta_1 (V_G - V_{s1} - V_{th})^2 = \mathrm{I_{M2}} = \beta_2 \left[ (V_G - V_{th}) V_{s2} - \frac{1}{2} V_{s2}{}^2 \right] \tag{2}$$

wherein $\beta_1$ and $\beta_2$ are the channel parameters of M1 and M2, respectively, $V_{th}$ is the threshold voltage of the transistors. From (2), we can calculate M1's transconductance and M2's drain-source resistance respectively, by:

$$\begin{cases} g_{m1} \cong \beta_1(V_G - V_{s1} - V_{th}) \\ r_{o2} = \frac{1}{\beta_2[V_G - V_{th} - V_{s2}]} \end{cases} \tag{3}$$

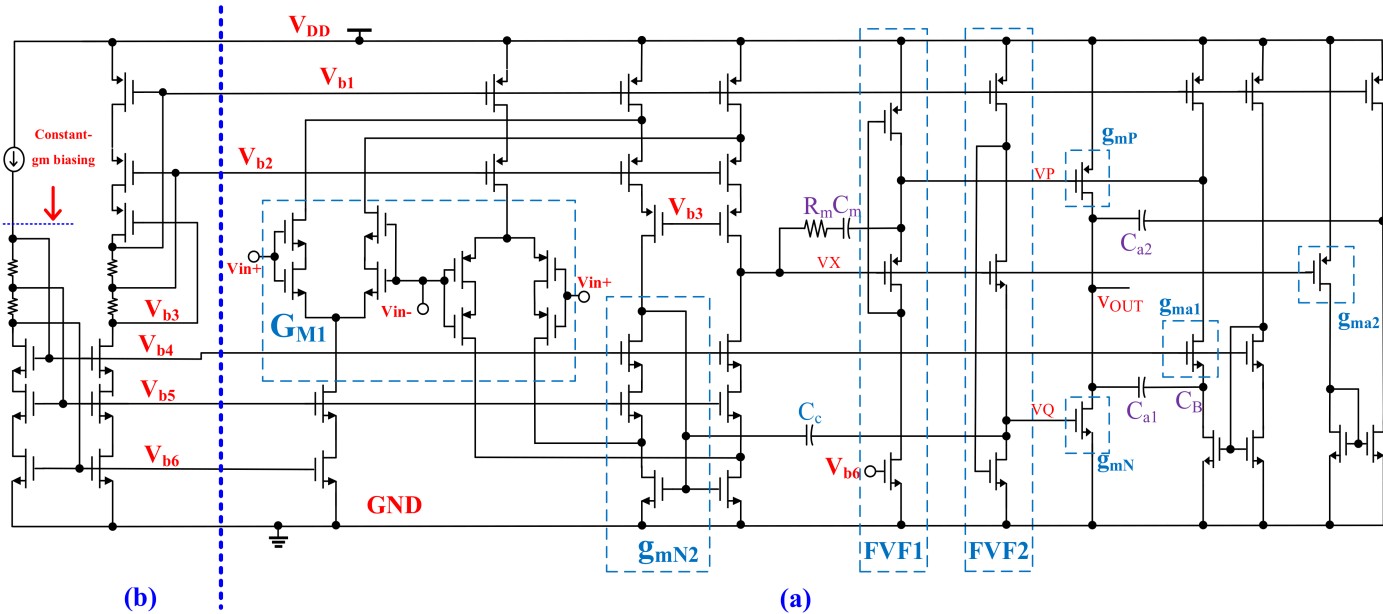

**Figure 1.** (**a**) The circuit implementation of the overall proposed driver amplifier and (**b**) the biasing circuits.

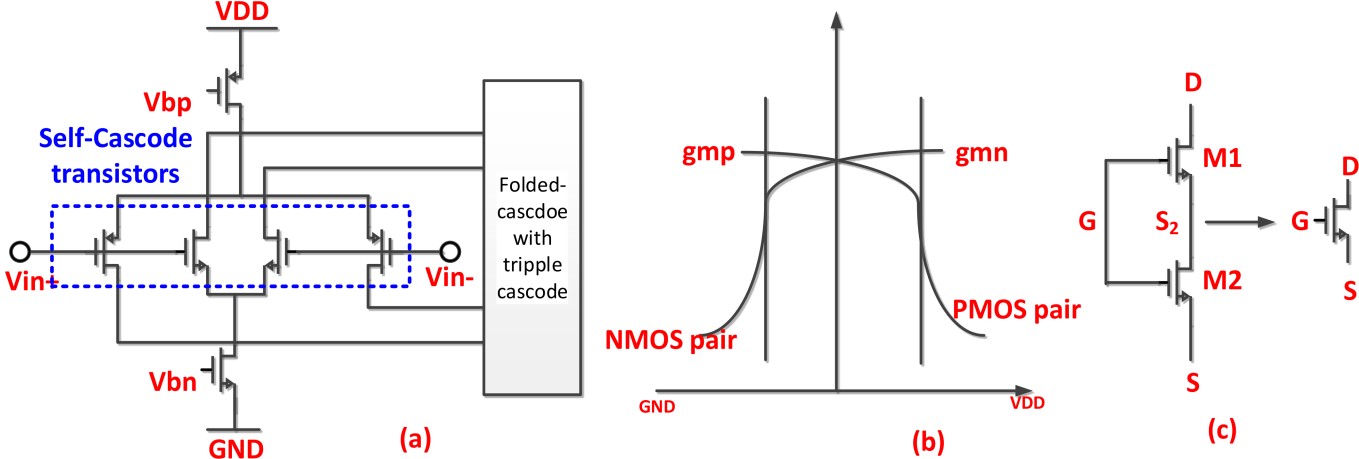

**Figure 2.** The rail-to-rail CMOS input stage with nMOS and pMOS differential pairs in parallel is adopted as the input stage: (**a**) Schematic and (**b**) the transconductance versus input common-mode range and (**c**) self-cascode input transistors.

Therefore, we can get that:

$$g_{m1}r_{o2} \approx \frac{(W/L)_1}{(W/L)_2} \tag{4}$$

wherein $(W/L)_2$ and $(W/L)_1$ are the channel width and length ratios of M2 and M1, respectively. The output equivalent resistance of the self-cascode transistors can be rewritten as:

$$r_{o,eq} = \left[ \frac{(W/L)_1}{(W/L)_2} + 1 \right] r_{o1} + r_{o2} \qquad (5)$$

As shown in Equation (5), the output resistance of self-cascode transistors can be increased by choosing a large $W/L$ for M2 or a small $W/L$ for M1.

### 2.2. The Flipped Voltage Follower (FVF) Buffer

The general purpose of the driver amplifier is to buffer output that follows the characteristics of the input signal under a large load while maintaining high THD suppression and low power consumption. The class-A or class-B output stage is not a viable option due to low power efficiency, output swing or high distortion. The push-pull class-AB output stage is a good alternative often used as an output stage in CMOS buffer amplifiers. The push-pull stage consists of a pair of complementary common-source transistors, allowing the rail-to-rail output voltage swing. The gates of the two output transistors are normally driven by two in-phase ac signals separated by a dc voltage.

The last class-AB stage should be constructed with large transistors to supply large current. Among many existing topologies, flipped voltage follower (FVF)-based buffers are an attractive topological choice to drive the output class-AB stage. The main advantage of the FVF is the reduced output resistance due to shunt feedback connection, which is the key for obtaining fast transient response and minimal area requirements for implementing amplifiers for on-chip applications. The schematic of a PMOS version of FVF is shown in Figure 3a, which consists of PMOS input transistor $M_1$, shunt feedback transistor $M_2$ and bias-current-source transistor $M_0$.

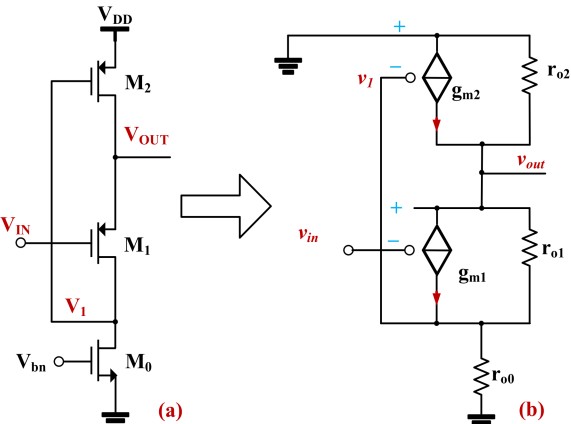

**Figure 3.** The voltage buffer topologies: (**a**) The flipped voltage follower buffer and (**b**) the small-signal model.

Figure 3b illustrates the equivalent small-signal model of FVF. According to the small-signal model, the output resistance of FVF buffer can be calculated as:

$$r_{FVF} = \frac{(r_{o0} + r_{o1})r_{o2}}{(1 + g_{m1}r_{o1})(1 + g_{m2}r_o)r_{o2} + r_{o1} + r_{o0}} \qquad (6)$$

wherein $g_{m1}$, $r_{o1}$ are M1's transconductance and output resistance respectively, $g_{m2}$, $r_{o2}$ are M2's transconductance and output resistance, respectively, and $r_{o0}$ is M0's output resistance.

It is reasonable to assume that

$$g_{m1}r_{o1} \gg 1, g_{m2}r_{o0} \gg 1$$

Therefore, Equation (6) can be simplified to:

$$r_{FVF} \approx \frac{1}{g_{m1}g_{m2}\frac{r_{o1}r_{o0}}{r_{o0}+r_{o1}}} \tag{7}$$

As shown by Equation (7), the output resistance of FVF can be greatly reduced by $g_{m1}(r_{o1}||r_{o0})$.

### 2.3. The Proposed Dual-Loop Active Frequency Compensation

Figure 4 shows the equivalent small signal model of the adopted DLAFC in this work. It consists of the two in-parallel signal paths (indicated by the blue dotted lines in Figure 4) and three active feedback paths (indicated by the red dotted lines, namely PMP and PMB). The amplifier's DC gain is dominated by the first input stage and the last output class-AB stage to achieve over 115 dB (refer to Figure 1). Therefore, the dominant pole $\omega_{don}$ occurred at the first stage while the nondominant pole $\omega_{non-don}$ contributed in the last output stage because of the large capacitance load. The most important issue in driver amplifier is to maintain high bandwidth and high DC gain, which can enhance the amplifier settling behavior and suppress THD, respectively. However, due to the class-AB output stage, that requires a large transistor size and large supply current for high slew rate and large output swing; it is very hard to move the nondominant location. Hence it is necessary to design a low-frequency LHP zero ($z_1 = -gm_{a2}/C_{a2}$) to compensate for the phase loss caused by the nondominant pole and achieve phase margin protection (PMP). Meanwhile, to achieve stability by mitigating the effect from the complex conjugate poles, two LHP zeros ($z_{2-3}$), one from $R_m$-$C_m$ pair and the other one from PMB ($z_3 = -gm_{a1}/(C_{a1} + C_B)$), are generated to compensate the phase margin. It is important to realize that to protect the THD suppression and achieve high-speed operation, the proposed DLAFC structure must prevent any direct connection between the Miller capacitor from the output gain stage to the first input gain stage. Therefore, the proposed DLAFC does not suffer from the bandwidth reduction caused by the Miller capacitive loading overhead at the amplifier output. The amplifier uses the proposed DLAFC compensation network with the following key characteristic components:

(1)  Capacitor $C_c$ is connected in the current buffer configuration in the inner loop, where its advantage is to shift the dominant pole location without any pole-splitting effect to protect the amplifier from PVT variations.

(2)  Phase Margin Boosting (PMB) is introduced to achieve nondominant pole compensation by creating 2 low-frequency LHP zeros from the PMB feedback path. A one-pair of Miller capacitor $C_m$ with nulling resistor $R_m$ in parallel with $G_{M,FVF1}$ is introduced to add 1-pole–zero pair to the inner loop effectively. The extra pole $\omega_{extra}$ (generated by capacitor $C_m$ with nulling resistor $R_m$ in the second stage) will not influence the amplifier stability because the AFFC network is connected between the FVF buffer stages and the class-AB output stage in cascode frequency compensation to generate pole splitting and to push this extra pole $\omega_{extra}$ into high-frequency pole $\omega_{HF}$ (Refer to the pole-zero placement relationship before and after compensation analyzed in the following sections along with systematic complete calculations ). AFFC topology is adopted to reduce the value of $C_c$ and push the dominant pole to higher frequency to obtain better THD+N suppression in higher frequency. The primary function of this RC pair is to create an extra zero along with the AFFC zero effectively: $z_3 = -gm_{a1}/(C_{a1} + C_B)$ to achieve phase margin boosting (PMB) against the complex poles.

(3)  Due to the phase margin loss from the second nondominant pole, it is necessary to introduce an extra LHP zero to achieve low-frequency phase margin protection (PMP). This is done through the PMP path, which effectively adds one more LHP zero ($z_1 = -gm_{a2}/C_{a2}$) to protect the phase margin.

The proposed DLAFC frequency compensation technique is capable of maintaining high gain (over 115 dB) at the high frequency when the amplifier is in a closed-loop configuration. To acquire the frequency property of the small-signal model in Figure 4, the analysis will start from the transfer function of the proposed three-stage amplifier, which can be characterized as:

$$T(s) = \frac{A_{DC}\left(1+\frac{s}{z_1}\right)\left(1+\frac{s}{z_2}\right)\left(1+\frac{s}{z_3}\right)}{\left(1+\frac{s}{\omega_{don}}\right)\left(1+\frac{s}{\omega_{non-don}}\right)\left(1+\frac{s}{Q\omega_0}+\frac{s^2}{\omega_0}\right)} \tag{8}$$

The transfer function is derived under the assumptions of the parasitical capacitances $C_1$, $C_2$ $C_3$ shown in Figure 4 that are much smaller than the frequency-compensation capacitances $C_c$, $C_m$, and the capacitance load $C_L$.

As previously mentioned, the extra pole $\omega_{extra}$ influence (generated by capacitor $C_m$ with nulling resistor $R_m$ in the second stage) along with the proposed LHP zero $z_2$ will be analyzed below with the model shown in Figure 5 before the complete analysis of the proposed DLAFC.

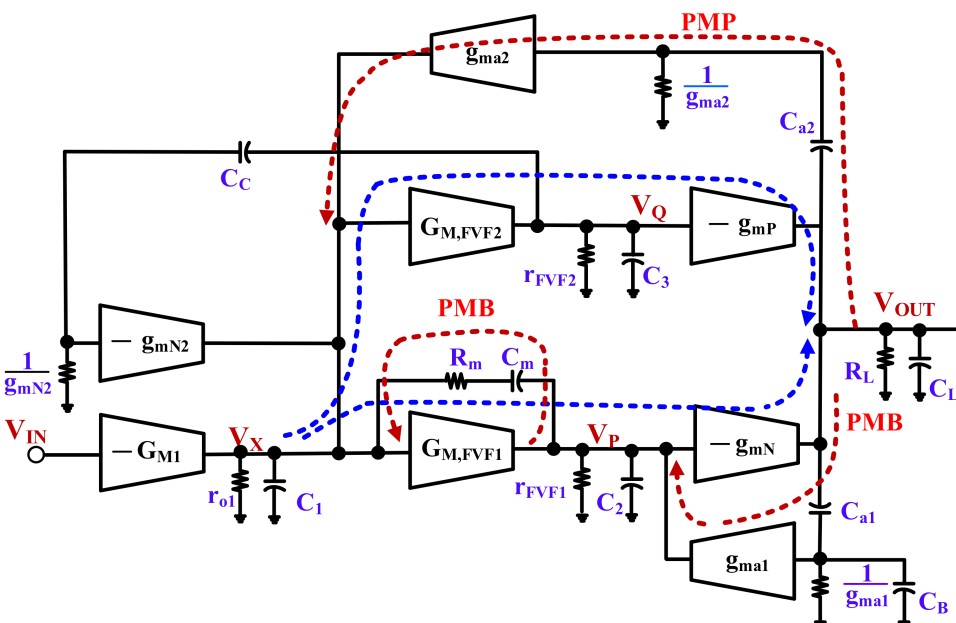

**Figure 4.** The detailed equivalent small signal model of the proposed DLAFC in this work.

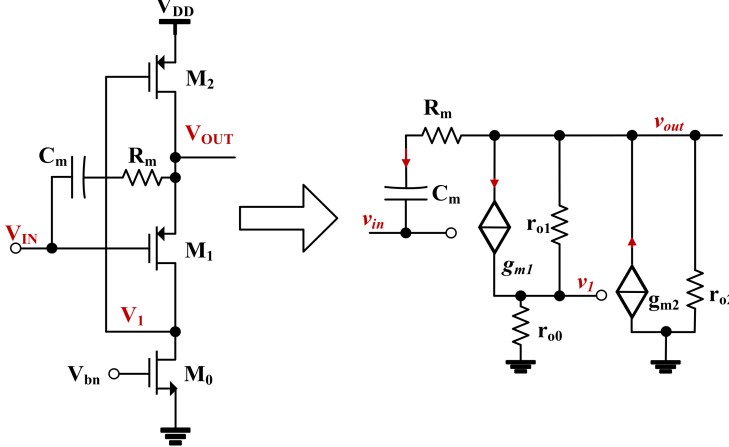

**Figure 5.** The equivalent small-signal model of the FVF buffer with the proposed $R_m$-$C_m$ pair for frequency compensation.

Based on the equivalent small-signal model shown in Figure 5, the following Equations can be derived:

$$\begin{cases} g_{m1}(v_{out} - v_{in}) + \dfrac{v_{out}-v_1}{r_{o1}} + \dfrac{v_{out}-v_{in}}{Rm+\frac{1}{sCm}} + \dfrac{v_{out}}{r_{o2}} = g_{m2}(0 - v_1) \\[2mm] \dfrac{v_{out}-v_1}{r_{o1}} + g_{m1}(v_{out} - v_{in}) = \dfrac{v_1}{r_{o0}} \end{cases} \tag{9}$$

From Equation (9), the transfer function of the FVF buffer can be expressed as:

$$\frac{v_{out}}{v_{in}} = \frac{g_{m1}r_{o1}g_{m2}r_{o2}r_{o0}}{g_{m1}r_{o1}g_{m2}r_{o2}r_{o0} + r_{o1} + r_{o0}} \frac{1 + sC_m\left(R_m + \frac{r_{o1}+r_{o0}}{g_{m2}g_{m1}r_{o1}r_{o0}}\right)}{1 + sC_m\left(R_m + \frac{r_{o1}+r_{o0}}{g_{m1}r_{o1}g_{m2}r_{o0}+\frac{r_{o1}+r_{o0}}{r_{o2}}}\right)} \tag{10}$$

Therefore, the compensation zero $z_2$ and the extra pole $\omega_{extra}$ can be calculated by:

$$z_2 = -\frac{1}{C_m\left(R_m + \frac{r_{o1}+r_{o0}}{g_{m2}g_{m1}r_{o1}r_{o0}}\right)} \approx -\frac{1}{C_m R_m} \tag{11}$$

$$\omega_{extra} = -\frac{1}{C_m\left(R_m + \frac{r_{o1}+r_{o0}}{g_{m1}r_{o1}g_{m2}r_{o0}+\frac{r_{o1}+r_{o0}}{r_{o2}}}\right)} \tag{12}$$

It is obvious that the $z_2$ and $\omega_{extra}$ are inevitably generated, and the $\omega_{extra}$ must be removed in order to make the frequency compensation effective, which allows the circuit only to take advantage of the effect of $z_2$ rather than being influenced by $\omega_{extra}$. Therefore, the cascode compensation by AFFC is used to generate pole-splitting effect to eliminate pole $\omega_{extra}$.

Secondly, it is necessary to analyze the frequency property of the AFFC cascade compensation shown in Figure 6. From the figure, we can obtain the following Equations:

$$\begin{cases} (i_{FVF1} + g_{ma1}v_{FB1})\dfrac{r_{FVF1}}{1+sC_2r_{FVF1}} = v_P \\[2mm] -g_{mP}v_P\dfrac{\frac{R_L}{1+sC_LR_L}\left(\frac{1}{g_{ma1}+sC_B}+\frac{1}{sC_{a1}}\right)}{\frac{R_L}{1+sC_LR_L}+\left(\frac{1}{g_{ma1}+sC_B}+\frac{1}{sC_{a1}}\right)} = v_{OUT} \\[2mm] v_{FB1} = \dfrac{sC_{a1}}{sC_{a1}+g_{ma1}+g_{ma1}+sC_B}v_{OUT} \end{cases} \tag{13}$$

wherein $v_P$ is the output voltage of buffer FVF1 while $v_{FB1}$ is the voltage feedback from the amplifier's output voltage $v_{OUT}$ through the AFFC capacitaor $C_{a1}$.$R_L$, $C_L$ are the amplifier's output resistance load and capacitance load, respectively. After combination and simplification of Equations in (13), we get:

$$v_P = -\frac{1}{g_{mP}}\frac{s^2C_{a1}C_LR_L + s(C_L + C_{a1})(g_{ma1} + sC_B)R_L + sC_{a1} + sC_B + g_{ma1}}{sC_{a1}R_L + sC_BR_L + g_{ma1}R_L}v_{OUT} \tag{14}$$

Substituting (14) in (13), we can get the transfer function of AFFC in the form of:

$$\frac{v_{OUT}}{i_{FVF1}} = -\frac{g_{mP}r_{FVF1}R_L\left(1 + s\frac{C_{a1}+C_B}{g_{ma1}}\right)}{HD(s)} \tag{15}$$

wherein the denominator of (15) is expressed as:

$$HD(s) = s^3\frac{C_{a1}}{g_{ma1}}C_LR_LC_2r_{FVF1} + s^2\frac{C_{a1}}{g_{ma1}}C_2r_{FVF1}\left(1 + \left(\frac{C_L}{C_{a1}} + 1\right)R_Lg_{ma1} + \frac{C_LR_L}{C_2r_{FVF1}}\right) + sC_{a1}r_{FVF1}R_Lg_{mP} \\ + s(C_L + C_{a1})R_L + s\frac{C_{a1}}{g_{ma1}} + sC_2r_{FVF1} + 1 \tag{16}$$

It is obvious that the AFFC can generate one low-frequency dominant LHP zero expressed as:

$$z_3 = -\frac{g_{ma1}}{C_{a1} + C_B} \tag{17}$$

Nondominant pole $\omega_{non-don}$ and high-frequency pole $\omega_{HF}$ by pole-splitting effect from AFFC shown in Figure 6 can be calculated from the derivations of the cascode frequency compensation as follows:

$$\omega_{non-don} = -\frac{1}{A_{v3}R_2 C_{a1}} = -\frac{1}{g_{m3} R_L r_{o2} C_{a1}} \tag{18}$$

$$\omega_{extra} ---> \omega_{HF} = -\frac{g_{m3} C_{a1}}{(C_{a1} + C_L) C_2} \tag{19}$$

wherein $g_{m3}$ is the transconductance of the last class-AB stage, which equals $(g_{mp} + g_{mn})$, $C_2$ is the total parasitical capacitance on FVF buffers' output. As implied by Equation (19), $\omega_{HF}$ can be in the GHz range when $g_{m3}$ is sufficiently large, and becomes a very high-frequency pole. Finally, the frequency property of the entire circuit can be summarized in Table 1.

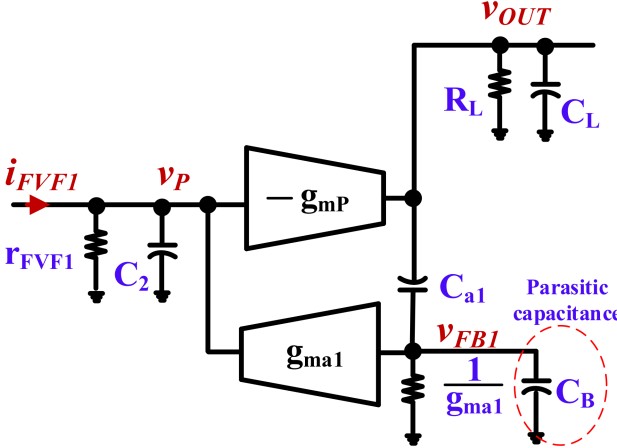

**Figure 6.** The equivalent small-signal model of the AFFC cascode compensation.

**Table 1.** The frequency property summarization of the proposed amplifier.

| Definition | Expression |
|---|---|
| DC gain | $A_{DC} = -G_{M1} r_{o1} g_{m3} R_L$ |
| Dominant pole | $\omega_{don} = -\frac{1}{g_{m2} R_2 (C_c + C_m) R_1}$ |
| Nondominant pole | $-\frac{1}{g_{m3} R_L r_{o2} C_{a1}}$ |
| High-frequency pole | $-\frac{g_{m3} C_{a1}}{(C_{a1} + C_L) C_2}$ |
| Low-frequency LHP zero | $z_1 = -\frac{g_{ma2}}{C_{a2}}$ |
| High-frequency LHP zero 1 | $z_2 = -\frac{1}{C_m R_m}$ |
| High-frequency LHP zero 2 | $z_3 = -\frac{g_{ma1}}{C_{a1} + C_B}$ |

The most advantage of the proposed technique compared with conventional simple Miller compensation (SMC), cascode frequency compensation (CFC), or nested/reversed nested Miller (NMC, RNMC) compensation techniques is that the proposed DLAFC is using the pole-splitting effect in an unconventional way between the second and third stage to only push the inevitable extra pole generated from the $R_m$-$C_m$ pair outside the gain-bandwidth product (GBW). Thanks to the positive phase shift provided by $z_{2-3}$, the stability of the DLAFC amplifier can still be achieved even when the GBW is set to be closer to $p_{1-2}$. Therefore, the GBW of the DLAFC amplifier can be improved by using the proposed pole-zero placement strategy, as shown in Figure 7, instead of using the third-order Butterworth response. As shown in Figure 8, the importance of the locations of the nondominant complex pole pairs gives rise to minor magnitude peaking. However, the effect on phase margin is mitigated by the presence of $z_2$ and $z_3$.

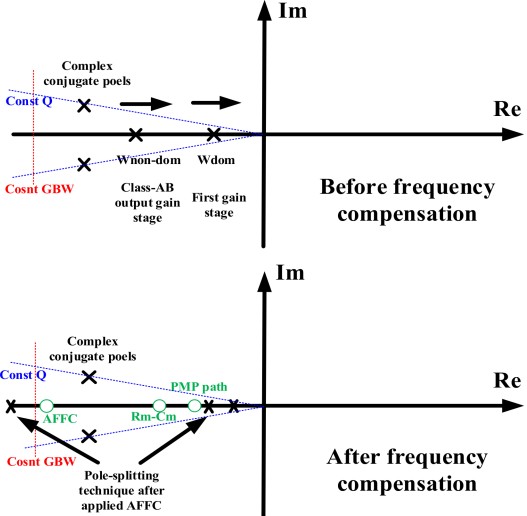

**Figure 7.** The pole-zero location diagram for illustrating frequency compensation.

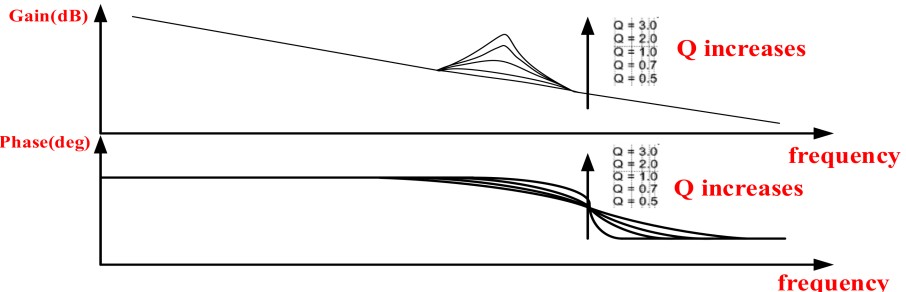

**Figure 8.** The frequency responses corresponding to a pair of complex poles with different Q values.

The designed complex conjugate poles implied by Equation (8) are given by:

$$p_{1-2} = \sqrt{\frac{gm_2 gm_3 C_{a2}}{C_{a1} C_L}} \tag{20}$$

And their phase-shifted amount can be calculated by:

$$\varphi(p_{1-2}) = \sum_{i=1}^{2} \arctan\left(\frac{\frac{\omega_{GBW}}{\omega_{cpi}}}{Q[1 - \left(\frac{\omega_{GBW}}{\omega_{cpi}}\right)^2]}\right) \tag{21}$$

wherein $\omega_{cpi}$ is the frequency of the designed complex conjugate poles.

The reason for designing the $Q$-value according to Equations (20)–(22) can be understood using Figure 8, which shows that the effect of a pair of nondominant complex poles can cause different $Q$-values on the voltage gain and phase shift of a generic three-stage amplifier. To avoid gain overshooting and dramatic phase reduction, as shown in Figure 8, $Q$-value of complex poles must be ensured to be smaller than 1. Therefore, the feature of having an additional positive phase shift due to two LHP zeros $z_{2-3}$ creates an opportunity to achieve stability by using an advanced pole-zero placement strategy. Due to the presence of two separate LHP zeros, the positive phase shift generated by $z_{2-3}$ will cancel out the amount of the negative phase shift caused by the nondominant complex poles $p_{1-2}$. The stability of the DLAFC amplifier is determined by its phase margin (PM) that is given as:

$$PM = 90 - \arctan\left(\frac{\omega_{GBW}}{\omega_{p2}}\right) - \sum_{i=1}^{2} \arctan\left(\frac{\frac{\omega_{GBW}}{\omega_{cpi}}}{Q[1 - \left(\frac{\omega_{GBW}}{\omega_{cpi}}\right)^2]}\right) + \sum_{i=1}^{3} \arctan\left(\frac{\omega_{GBW}}{\omega_{zi}}\right) \quad (22)$$

wherein $\omega_{p2}$ is the nondominant pole and $\omega_{zi}$ is the frequency of the $i$-th zero.

### 3. Simulation Results and Discussions

To verify the functionality of the proposed DLAFC-based scheme driver amplifier design, the proposed prototype is implemented in the 0.18 μm DBH 5V technology process. The amplifier is expecting to drive a maximum load of less than 60 pF, including the probe capacitance. The first dominant pole is pushed around 5 KHz to contain enough loop gain to suppress THD. Figure 9 shows the frequency responses of the proposed amplifier with the loading of 10 pF, and the poles and zeros from the previous analysis are labeled on the plot. It shows that under the proposed delicate pole-zero placement, the phase margin of a multistage amplifier can be optimized to approximately equal 90°; the nominal GBW is around 250 MHz.

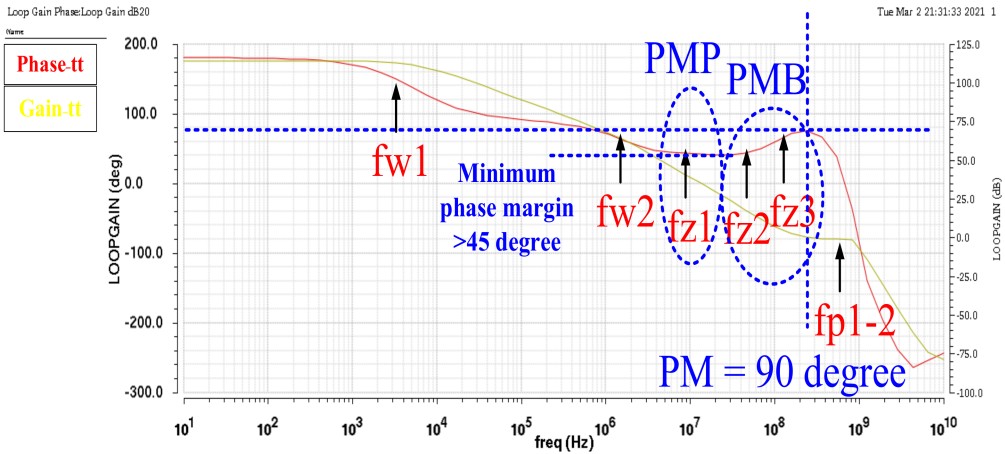

**Figure 9.** The simulated bode diagrams of the DLAFC configuration.

Figure 10 shows that the proposed diver amplifier can achieve a robust operation under a wide range of load capacitance. The simulated results for the proposed DLAFC amplifier are further verified with different capacitive loads to prove the robustness of the proposed design. Figures 11 and 12 show the proposed amplifier can achieve slew rate SR+- greater than 250 V/μs under small voltage swing and large voltage swing. The SR test can also be a reasonable verification of the robustness of the proposed frequency compensation technique in this work.

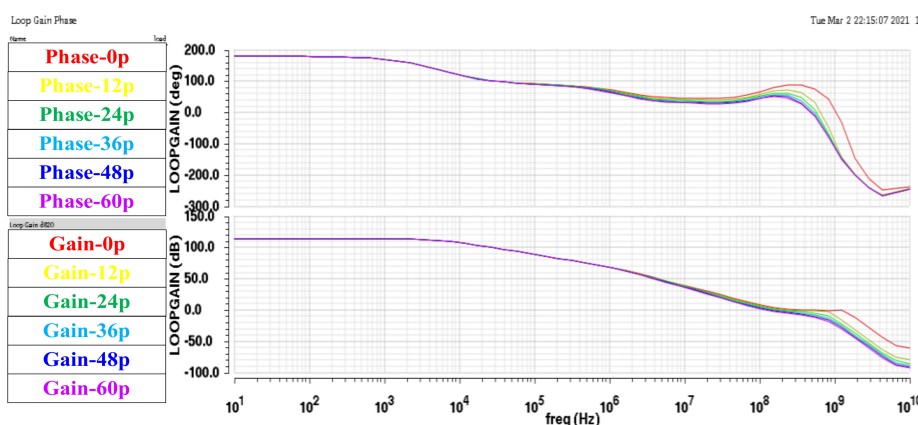

**Figure 10.** The simulated bode diagrams of the DLAFC configuration under different capacitor load between 0–60 pF.

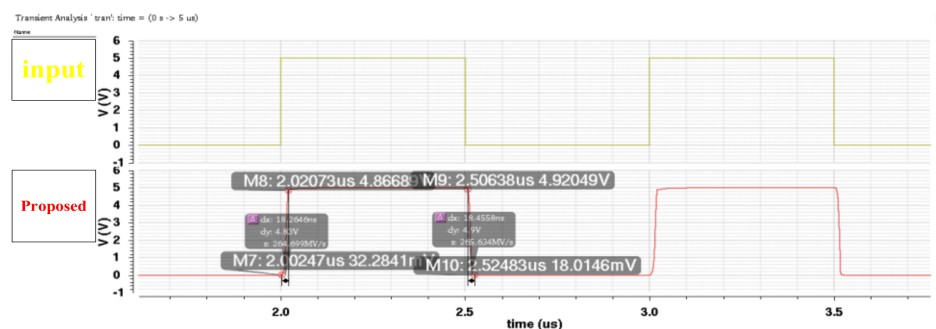

**Figure 11.** The transient simulation results of slew rate when the proposed circuit in closed-loop state is fed with a large-swing input.

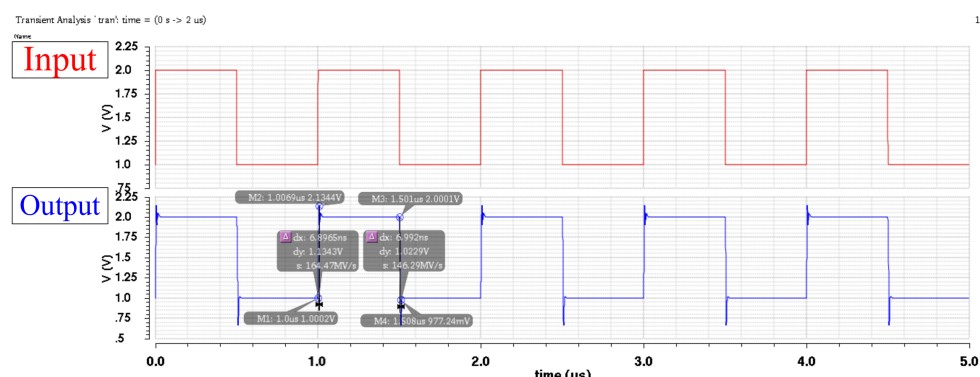

**Figure 12.** The transient simulation results of slew rate when the proposed circuit in closed-loop state is fed with a small-swing input.

Figure 13 provides the frequency response under process variations, and it is clearly showing the proposed DLAFC is robust under most extreme corners. Figures 14 and 15 show that the proposed amplifier can achieve THD suppression at high-frequency operation (@1 MHz and 8 MHz) under large voltage swings 1 Vpp and 2 Vpp.

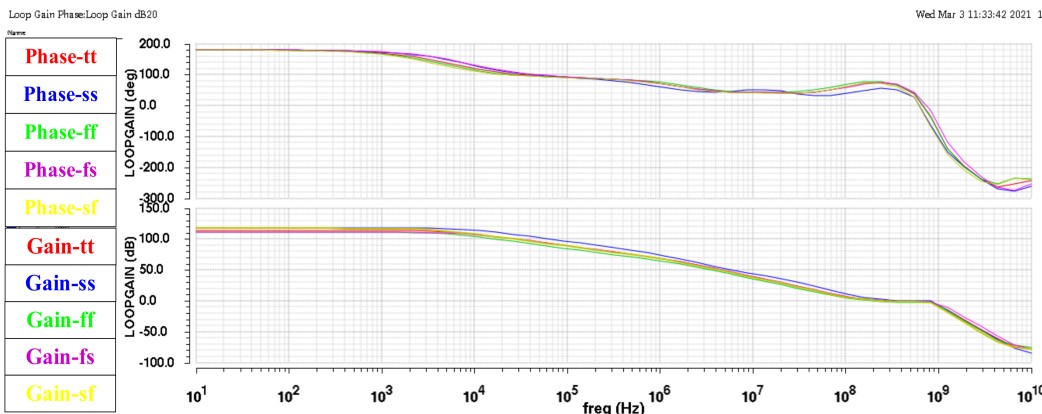

**Figure 13.** The simulated bode diagrams of the DLAFC configuration under 5 different process corners.

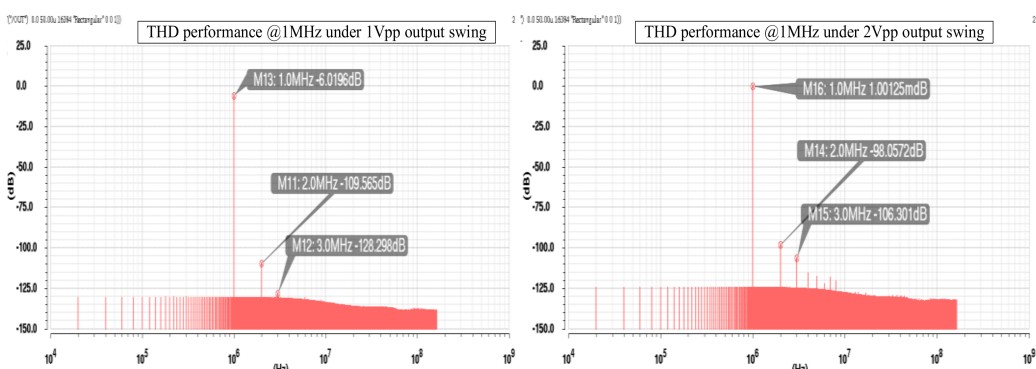

**Figure 14.** The simulation results of the total harmonic distortion of the proposed circuit under 1 V and 2 V output swing operating at 1 MHz.

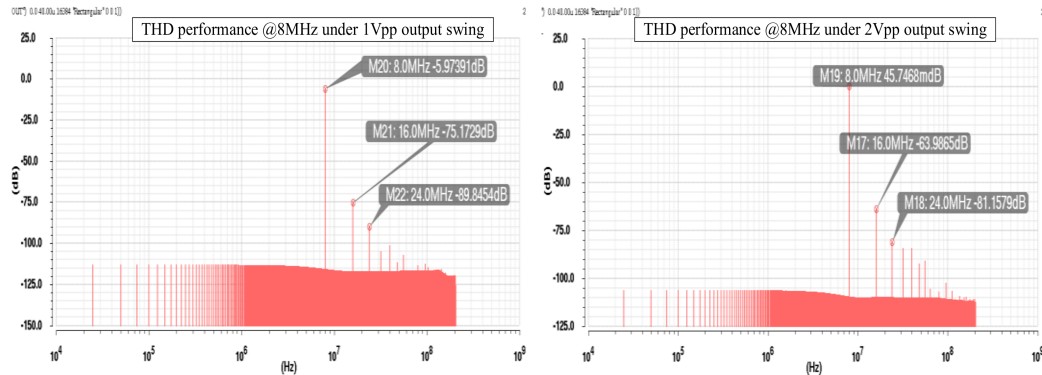

**Figure 15.** The simulation results of the total harmonic distortion of the proposed circuit under 1 V and 2 V output swing operating at 8 MHz.

Figures 16 and 17 show the excellent performance of the input-referred noise, PSRR, and CMRR, output impedance under all different process corners to demonstrate the proposed circuits can maintain high immunity to power-supply noise and common-mode noise, and maintain a low noise floor under high-frequency operation. The ultra-low output impedance (less than 10 ohms) within 100 MHz shows the proposed amplifier can have a very powerful driving capacity without suffering from signal distortion. Overall, it is clear that the proposed design is robust in terms of the circuit's architecture, THD suppression, and frequency compensation scheme.

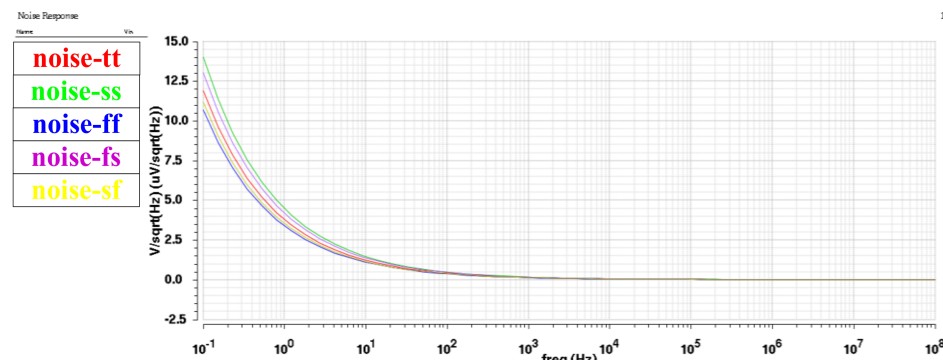

**Figure 16.** The simulated noise power-density spectrum of the proposed circuit under 5 different process corners.

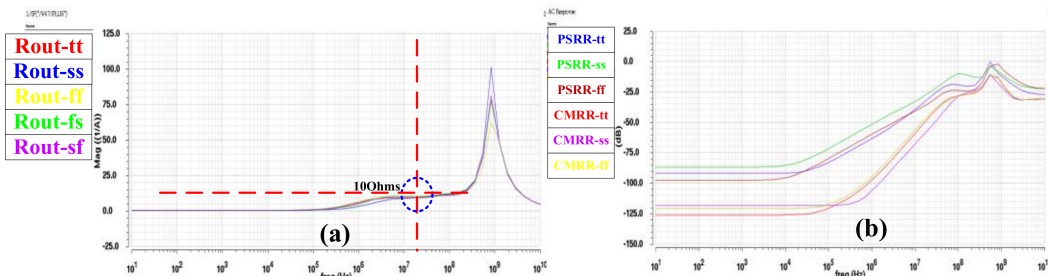

**Figure 17.** The simulation results of output impedance, power-supply rejection ratio (PSRR), and common-mode rejection ratio (CMRR). (**a**) The output impedance (**b**) PSRR & CMRR.

The performance of this work is summarized in Table 2 and is compared with the state-of-art. As shown in the table, the proposed design can achieve a very high DC gain, a high GBW, a small input offset, and high THD suppressions simultaneously while its power consumption is relatively low compared with others, which proves that our design offers a very good bandwidth-power efficiency.

**Table 2.** The performance and comparison to the prior works.

| Parameter | [1] This Work | ** [3] | ** [4] | ** [5] | ** [6] |
|---|---|---|---|---|---|
| Technology | **DBH-0.18** μm | Bipolar/BICMOS | Bipolar/BICMOS | Bipolar/BICMOS | 0.18 μm BICMOS |
| Supply | **5** | 5 | 5 | 5 | 3.3 |
| Power (mW) | **12.5 mW** | 5.7 mW | 5 mW | 11 mW | 120.12 mW |
| A0 (dB) | **115** | 70 | 45 | 33 | 97 |
| GBW (MHz) | **250** | 50 | 34 | 30 | 2200 |
| SR (V/μs) | **265** | 490 | 45 | 22 | |
| Input referred noise | **5 nV/$\sqrt{\text{Hz}}$ @1 MHz** | 5.7 nV/$\sqrt{\text{Hz}}$ @1 MHz | 6.4 nV/$\sqrt{\text{Hz}}$ @1 MHz | 5.1 nV/$\sqrt{\text{Hz}}$ @1 MHz | 1.67 nV/$\sqrt{\text{Hz}}$ @1 MHz |
| HD2,3 @ (1 MHz output swing (1 V, 2 V) | **16.5-bit 16-bit ($-109/128$) ($-98/106$)** | HD2,3 at 10 kHz (dB) $-133/140$ | HD2,3 at 10 kHz (dB) $-106/-103$ | HD2,3 at 10 kHz (dB) $-125/-126$ | HD2,3 at 10 kHz (dB) $-74/-101$ |
| HD2,3 @ (8 MHz) output swing (1 V, 2 V) | **14-bit 13.5-bit ($-75/-89$) ($-63/-81$)** | | | | |
| Input offset (μV) | **85** | 2000 | 200 | 360 | ---------- |

[1] Simulation results, ** others are from state-of-art products and publications.

## 4. Conclusions

In this paper, a DLAFC-based driver amplifier is presented. The theoretical analysis and simulation results show good agreement regarding excellent DC gain, GBW, speed, input-referred noise and THD, which is suitable for high-speed applications. Moreover, the proposed amplifier is proven stable in most extreme PVT variations to validate the design robustness. The proposed amplifier has competitive performance compared with state-of-art amplifiers used as products.

**Author Contributions:** Data curation, X.F.; Formal analysis, X.F.; Funding acquisition, Y.Y.; Methodology, X.F.; Project administration, Y.Y.; Supervision, K.E.-S. and Y.Y.; Validation, X.F.; Writing—original draft, X.F.; Writing—review & editing, Y.Y. and K.E.-S. All authors have read and agreed to the published version of the manuscript.

**Funding:** This research was partly funded by the Fujian Provincial International Cooperation Projects of China, grant number 2020I0005, and The Fujian Provincial Industry-Academia-Research Cooperation Projects of China, grant number 2020Y4017.

**Conflicts of Interest:** The authors declare no conflict of interest.

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
