# Peer review of "A High Bandwidth-Power Efficiency, Low THD2,3 Driver Amplifier with Dual-Loop Active Frequency Compensation for High-Speed Applications"

_electronics, doi:10.3390/electronics10182311_

Round 1

Reviewer 1 Report

Major comments and suggestions:

  1. Please rewrite the Abstract in a concise manner.

  1. There are a lot of sentences with miscellaneous grammar issues needed to be rewritten. Examples include Lines 59-60, 81-83, 133-134, 141-144, 151-155, 226-229, and 255-257, etc. Please also avoid using too long/complicated sentences for the ease of reading.

  1. Figure 1 is in need of elaboration. The connection between each part of it and the corresponding subsection in Section II should be clearly pointed out as well.

  1. Please be consistent when referring to a figure (Fig. # or Figure #).

  1. The subfigures of Figure 4 should share one caption.

  1. Some apparent steps of derivation for Equation (13) can be reduced.

  1. Variables in Equations (24)-(26) need to be explained.

  1. Figures 9-17 should be elaborated with more descriptions and analyses.

  1. Table 1 has never been referred to or analyzed.

Author Response

Sincerely

Ximing

Reviewer 2 Report

A research article (manuscript ID: electronics -1361140) entitled “A High Bandwidth-Power Efficiency, Low THD 2,3 Driver Amplifier with Dual-Loop Active Frequency Compensation for High-Speed Applications” by an international team from Canada and China was submitted to the MDPI Electronics Journal.

This paper has 14 pages including 17 figures, 1 table, and 15 references. In their investigations, the authors present a driver amplifier suitable for integration in 16-18-bit high-speed ADCs, liquid crystal display (LCD) drivers, or any other similar applications. This paper can be interesting for researchers and engineers as well as for graduate and postgraduate students studying this subject.

The reviewer has looked through the paper and found that this presentation requires some revision.

For instance, the following correction of the English language can be done in the paper.

1)  line 61: “nondominant” instead of “non-dominant”;

2)  line 74: “Figure 1” instead of “Figure.1”;

3)  line 76: “Figure 2a” instead of “Figure. 2(a)”;

4)  line 87: “Figure 2” instead of “Fig. 2”;

5)  line 89: probably “M1 (VS2-VS in Figure 2b) is” instead of “M1 in Fig.2 (b) VS2-VS is”;

6)  line 113: “Figure 3b” instead of “Fig. 3(b)”;

7)  line 130: “Figure 4a” instead of “Figure. 4 (a)”;

8)  line 132: “115 dB (Figure 1). In the” instead of “115 dB.(refer to Fig.1) In the”;

9)  lines 133, 139, 158: “nondominant” instead of “non-dominant”;

10)  lines 162, 167: “shown in Figure 4b” instead of “in Figure.4(b)”;

11)  line 179: “(Figure 5)” instead of “of Figure. 5”;

12)  lines 172-174: This long sentence should be improved and simplified (use two or moreshort sentences instead of one large one);

13)  line 182: add to the end “. Therefore, ” ;

14)  line 259: “Figures 11 and 12” instead of “Fig.11 and Fig. 12”;

15)  line 261: “Figure 13” instead of “Fig.13”;

16)  line 253: “60 pF” instead of “60pF”;

17)  line 254: “Figure 9 shows” instead of “Figure. 9 shows”;

18)  line 257: “250 MHz. Figure 10 shows” instead of “250MHz. Fig.10 shows”;

19)  line 263: “Figures 14 and 15 show” instead of “Fig. 14 and Fig.15 shows”;

20) line 264: “Figures 16 and 17 show” instead of “Figure 16 and Figure 17 show”;

Etc.

The formulas between formulas (12) and (13) must be improved: use r02 instead of r02, etc. Vin and vin are the same?

In formula 27, do not use Italic for arctan.

Also, it looks like many formulas contain some parameters that were not explained in the context.

Please, use always “The” in any figure (or table) title after each figure (table) number. Also, use a period at the end of each figure title. For instance:

Table 1. The performance table” instead of “Table 1. Performance table”;

In the table, please, use the minus sign “–” instead of “-” for dB.

Figure 1. (a) The circuit implementation of the overall proposed driver amplifier and (b) the biasing circuits.” instead of “Figure 1. (a) Circuit implementation of the overall proposed driver amplifier and (b) biasing circuits.”;

Figure 2. The rail-to-rail CMOS input stage with nMOS and pMOS differential pairs in parallel is adopted as input stage: (a) Schematic and (b) the transconductance versus input common-mode range and self-cascode input transistors.” instead of “Figure 2. Rail-to-rail CMOS input stage with nMOS and pMOS differential pairs in parallel is adopted as input stage (a) Schematic. (b) Transconductance versus input common-mode range & self-cascode input transistors”;

Figure 3. The voltage buffer topologies: (a) The flipped voltage follower buffer and (b) the small-signal model.” instead of “Figure 3. Voltage buffer topologies (a) flipped voltage follower buffer (b) small-signal model”;

Figure 4a. The structure” instead of “Figure 4a. Structure”;

Figure 4b. The equivalent” instead of “Figure 4b. Equivalent”;

Figure 5. The equivalent” instead of “Figure 5. Equivalent”;

ETC. Please, use a period always after each figure title.

The English language must be polished. The corresponding figures and table titles must be improved. The table must be improved. The references must be in the MDPI format. The paper requires a major revision.

Round 2

Reviewer 1 Report

Most of the previous comments and concerns have been addressed and the manuscript is in a much better shape. I appreciate the authors’ detailed responses.

Author Response

Dear Electronics Assigned Editor

September-13th

Dear Editors & Reviewers:

Please find below our second reply to the reviewers’ comments inserted in red color. We have provided a line-by-line response with ultimate consideration, and we have revised our paper, especially in the English editing, according to the reviewers’ comments. 

We know no conflicts of interest associated with this publication. As the first author, I confirm that the manuscript has been read and approved for submission by all named authors.

Thank you for your consideration of this manuscript.

Sincerely

Ximing Fu

Department of Electrical and Computer Engineering,

Dalhousie University,

Halifax, NS B3H 4R2, Canada

(e-mail: [email protected])

Open Review

English language and style

( ) Extensive editing of English language and style required
( ) Moderate English changes required
(x) English language and style are fine/minor spell check required
( ) I don't feel qualified to judge about the English language and style

Yes

Can be improved

Must be improved

Not applicable

Does the introduction provide sufficient background and include all relevant references?

( )

(x)

( )

( )

Is the research design appropriate?

( )

(x)

( )

( )

Are the methods adequately described?

(x)

( )

( )

( )

Are the results clearly presented?

( )

(x)

( )

( )

Are the conclusions supported by the results?

( )

(x)

( )

( )

Comments and Suggestions for Authors

Most of the previous comments and concerns have been addressed, and the manuscript is in much better shape. I appreciate the authors’ detailed responses.

A: Thank you very much for reviewer1’s encouragement. We will provide English editing with ultimate consideration in the newly revised manuscript.

Submission Date

14 August 2021

Date of this review

14 Sep 2021 01:25:03

Reviewer 2 Report

A research article (revised manuscript ID: electronics -1361140) entitled “A High Bandwidth-Power Efficiency, Low THD 2,3 Driver Amplifier with Dual-Loop Active Frequency Compensation for High-Speed Applications” by an international team from Canada and China was submitted to the MDPI Electronics Journal.

The revised paper has 15 pages including 17 figures, 2 tableы, and 15 references. In their investigations, the authors present a driver amplifier suitable for integration in 16-18-bit high-speed ADCs, liquid crystal display (LCD) drivers, or any other similar applications. This paper can be interesting for researchers and engineers as well as for graduate and postgraduate students studying this subject.

It is necessary to state that tyhe authors have done many changes and improvements in their paper. However , this paper still requires a major revision because all the references should be in the MDPI format and the English language mus be significantly improved.

For instance, line 300 : "In this paper, a DLAFC based driver amplifier is presented. " instead of "In this paper, a DLAFC based driver amplifier is presented in this paper. ";

In the table 2 title: "The performance" instead of "The Performance";

Also, in Table 2, the second line, second column: "µm" instead of "um";

the seventh line, first column: "µs" instead of "us";

the last lene, first column: "µV" instead of "uV";

Also, in the tables and in the paper text, all the physical dimesions should be separated from the numbers by a space.

Line 245: "µm" instead of "um";

Line 242 is unclear!

Line 232, please, use "×" instead of "*";

Line 175: "shown in Figure 5" instead of "in Figure5";

Line 171: "will be analyzed below with the model shown in Figure 5" instead of "will be analyzed from Figure 5 in the following";

Line 168: "capacitances" instead of "capacitance" and " shown in Figure 4 that are" instead of " shown in Figure 4, are";

etc. There are too many such places that should be corrected and inproved concerning the English language.

Author Response

Dear Electronics Assigned Editor

September-13th

Dear Editors & Reviewers:

Please find below our second reply to the reviewers’ comments insert in red color. We have provided a line-by-line response with ultimate consideration and we have revised our paper especially in the English editing according to the reviewers’ comments. 

We know no conflicts of interest associated with this publication, As the first author, I confirm that the manuscript has been read and approved for submission by all named authors.

Thank you for your consideration of this manuscript.

Sincerely

Ximing Fu

Department of Electrical and Computer Engineering,

Dalhousie University,

Halifax, NS B3H 4R2, Canada

(e-mail: [email protected])

Round 3

Reviewer 2 Report

A research article (second revision, manuscript ID: electronics -1361140) entitled “A High Bandwidth-Power Efficiency, Low THD 2,3 Driver Amplifier with Dual-Loop Active Frequency Compensation for High-Speed Applications” by an international team from Canada and China was submitted to the MDPI Electronics Journal.

This paper (second revision) has 15 pages including 17 figures, 2 tables, and 15 references. In their investigations, the authors present a driver amplifier suitable for integration in 16-18-bit high-speed ADCs, liquid crystal display (LCD) drivers, or any other similar applications. This paper can be interesting for researchers and engineers as well as for graduate and postgraduate students studying this subject.

The authors have done many improvements in the second revision of the paper.  However, it is necessary to use the MDPI format for the References and on lines 241, 231, 210, 200, 194 it must be written "where" instead of "Where in" or "Wherein".